# The Question of Beauty and the Aesthetic Value of the Image of the Mother of God in Pastoral Care and Catechesis

**Mateja Pevec Rozman \* and Tadej Strehovec \***

Faculty of Theology, University of Ljubljana, 1000 Ljubljana, Slovenia
\* Correspondence: mateja.pevec@teof.uni-lj.si (M.P.R.); tadej.strehovec@teof.uni-lj.si (T.S.)

**Abstract:** Ancient philosophers attached great importance to the ideals of unity, truth, goodness, and beauty as the path to the greatest good. Beauty expressed through works of art can open the eyes of the mind and heart and direct the human spirit to transcendence. The beauty of art awakens inner emotionality, evokes elation in silence, and leads to "coming out of oneself". The concept of beauty is inextricably linked to Catholic theology and preaching. Beautiful sacred images were a source of theological messages, intercession, and entry into the transcendental world. Medieval Gothic cathedrals had images on the walls as a basic tool of catechesis. Even in today's teaching of young people, images play a crucial role. The world of symbols enables Christians to connect everyday life experiences with theological messages. The image of the Virgin Mary is the best example of recognizing personal life situations, with the story of a mother who loved her child, accepted the suffering and death of her own son, and together with St. Joseph formed a holy family, which is the image of an imperfect family in which each person recognizes himself. All these aspects of the life of the Virgin Mary could form the basic concepts of the Christian understanding of beauty. In modern thought, the concept of beauty is understood quite narrowly (we are talking about narrowing the meaning of the concept of beauty). In the first part of this paper, we focus on the philosophical concept of beauty with a brief historical overview, then we point out the difference between transcendental beauty and aesthetic beauty. Beauty appeals to the human being and opens the heart to the transcendent, to God, who is the source and fullness of beauty, beauty itself. The originality of the article is in its presentation of the understanding of the Christian concept of beauty through the figure and image of Mary, the Mother of God. The experience of the beauty of Mary and Mary's life story enables the believer to have a different perspective on the perception of his own life and thereby opens him to the transcendent, to a personal relationship with God, who is eternal Beauty.

**Keywords:** beauty; transcendental beauty; aesthetics; sacred art; catechesis; Virgin Mary; Christianity; pastoral care; evangelization

## 1. Introduction

The beauty of Mary is a concept for understanding various theological perspectives of Mary's life, through which one can reflect on one's own life situation and enter into the world of the transcendent. The ultimate purpose of Mary's beauty is to help a human person to deepen his personal faith and to strengthen his belonging to God. To understand Mary's beauty, it is necessary to place the understanding of beauty in the context of ancient and medieval philosophy and to present the general understandings of beauty in modern Catholic theology. The understanding of beauty in modern times is not always uniform, its meaning has changed over time and has acquired different meanings. Therefore, the first part of the article is devoted to the philosophical and theological understanding of the concept of beauty. In the second part of the article, the understanding of the concept of beauty is extended to the concept of the beauty of Mary as a part of the so-called *via pulchritudinis*. The definition of the Mary's beauty does not consist in a general presentation

and analysis of Mariology, but in placing the Mary's beauty in the context of philosophical–theological reflection, and thus in pointing out its practical catechetical usefulness as a means of transmitting a religious message. The existential and theological dimensions of Mary's beauty reveal the dimensions of Mary's concrete life and enable humans to be open to transcendence and to establish a personal relationship with God.

## 2. The Concept of Beauty

The question of beauty is a question which has occupied humans from the very beginning of history, from the beginnings of self-awareness and the awareness of the difference between the self and the other, the difference between humans and the outside world. "Enigmatic and enchanting, persistent and irresistible, beauty, as a concept, fascinates and accompanies human thought through time, given rise to inspiration and emotion, dialogue, search, exploration and analysis, juxtaposition and objection" (Lagogianni-Georgakarakos 2018, p. 35). The concept of beauty occupied the interest of the first philosophers, who wanted to find the essence of beauty, and artists, who tried to find the way to translate this "incomprehensible something into shape, color, form, image, symbol, idea, or value leaving an imprint on human creation with its continuous variants and countless aspects" (35). Beauty is something excellent, something that attracts, that evokes admiration, pleasure, fullness of feeling, and satisfaction.

### 2.1. Philosophical Approach

"What is this, the beautiful?" Socrates wondered (Plato 1925, Hippias Major, 287d). Is beautiful something that exists? A being? A character? A deed? Is beauty something appropriate (293d–295a), something beneficial, useful (295a–296d), a pleasure that arises from eyesight and hearing (297d–303d)?

The Cambridge Dictionary of Philosophy offers this definition: "Beauty, an aesthetic property commonly thought of as a species of aesthetic value. As such, it has been variously thought to be (1) a simple, indefinable property that cannot be defined in terms of any other properties; (2) a property or set of properties of an object that makes the object capable of producing a certain sort of pleasurable experience in any suitable perceiver; or (3) whatever produces a particular sort of pleasurable experience, even though what produces the experience may vary from individual to individual. It is in this last sense that beauty is through to be "in the eye of the beholder" (Audi 1999, s.v. "Beauty" in: The Cambridge Dictionary of Beauty). According to this description we see that it is not possible to establish a single definition that would cover all aspects of the concept of beauty. Even Socrates in his dialogue with Hippias[1] does not offer a clear answer: The question of the beautiful is not adequately answered in the dialogue Hippias Major (303d–304e).

The Greek concept of beauty itself, *kalón*[2] (Latin *pulchrum*), does not refer so much to things that have an autonomous aesthetic value, but rather to excellence, which is linked to the concept of moral excellence or has a moral connotation. This concept is closer to Kant's notion of "*dependent beauty*, possessed by an object judged as a particular kind of thing (such as a beautiful horse) than it is to free beauty, possessed by an object judged simply on the basis of its appearance and not in terms of any concept of use" (Audi, s.v. "Beauty" in: The Cambridge Dictionary of Beauty).

If we start in ancient Greece, the exploration of beauty began about twenty-seven centuries ago. For the Greeks, the concept of *kalón* included various meanings: beautiful as attractiveness (an aesthetic aspect) or as something worth admiring, beautiful as utility (as something that satisfies human needs), and beautiful as moral rightness (Klun 2002, p. 100). Ancient Greeks held a fascination for the beauty of the body; they admired the beauty of the human body, and the body became an important subject of artistic endeavor for them. They attributed to the gods a human image, an image of ideal proportions, to which humans must aspire. The human body is at once sacred and secular, "a system" that operates according to precise laws. This is particularly pronounced in the classical

period of Greek art, especially sculpture, when sculptures become increasingly naturalistic (pronounced anatomy in marble) (Pevec Rozman 2022, p. 15).

*2.2. Short Historical Overview (from Ancient Times until the Middle Ages)*

Already the Pre-Socratics were interested in defining the concept of beauty, first with the concepts of symmetry and harmony. Pythagoras (ca. 580–500 BC) taught the beauty of mathematics and argued that numbers and the harmony which they entail express with remarkable precision the essence of beings and phenomena (Turner 1903). Heraclitus (ca. 535–475) talked about the cosmic harmony between opposites and proclaimed that to God all things are beautiful and good and just (Lagogianni-Georgakarakos 2018, p. 36). Until Plato, Greek thought remained mainly on the descriptive and aesthetic level, on the level of the beautiful, which is pleasing to the eye and was often associated with erotic desire (Stres 2018, s.v. Beauty). A systematic analysis of beauty thus began only in Athens during the 4th century BC, by the leading figures of Attic philosophy. Plato stressed the inadequacy of the current Greek view, which was primarily concerned with the aesthetic aspect, with the beauty of the body.

According to Plato (427–347 BC), humans must ascend (outshine) the beauty of things to grasp beauty itself, the beauty which is the origin of all the beauty (Platon 2004b, Symposium, 210e–211c). The beautiful, or the idea of beauty, belong to the world of ideas. Beauty is self-existent; it has unified form and is eternal (Platon 2004b, Symposium, 211a; Platon 2004a, VI, 507b). Even if in Socratic dialogue Hippias Major the question "What is beautiful?" does not receive a clear answer, "it is emphasized that the beautiful belongs to the world of ideas, confuting the prevailing views" (Giannopoulou 2018, p. 71). Beauty is something that has a supersensible and transcendent origin, and if one does not stop at mere passing beauty, it also has the attractive power to lead one into the realm of divine ideas, where one is at home (Phaidon, 250c–251a). In *Republic*, Plato described the artist who is practicing imitation (gr. *mimesis*). The artist, who produces an image of an object (for example, a bed), imitates (produces) a particular copy of this object (bed). This object for Plato is only a pale imitation of the eternal Form of this object (Cottingham 2008, p. 695).

Aristotle (384–322 BC), who is considered a realist (in the modern sense of the word), as opposed to Plato, who is considered an idealist and a spiritualist (Kos 1970, p. 37), after the first faithful steps of following his teacher, turned away from his theory of supersensible ideas and devoted himself more to the question of the search for the origin, the foundation in this world, in the material world. He defines the human being as a composite being (body and soul), as a rational being who is also by nature a social being, and "fulfilment is attained through rational thought and virtue" (Giannopoulou 2018, p. 75). Aristotle denies, as diametrically opposed to the theory of ideas and the ascetic strictness of Plato, the distinction between the intellectual and the sensible world and searches for the objective law that governs beauty and happiness in the tangible world and not in a transcendental field (75). Aristotle is convinced that beauty is an objectively existing quality, "an attribute things themselves are marked by, that becomes accessible through the laws of reasoning and methodical classification, but also the balance of passions and emotions" (Giannopoulou 2018, p. 75). As was typical of ancient aesthetics, Aristotle (in his *Metaphysics*) stressed that beauty requires orderliness, symmetry (in space), proportion and clarity, proportionality, and coherence, later adding (in his *Poetics*) the virtues of integrity and unity and multiformity. Aristotle adds magnificence and grandeur to the conditions for fulfilling the criteria of the beautiful. For Aristotle, the only one who possesses perfect beauty, the only one who is unchangeably perfect, beautiful, and good, is the "first mover" (Unmoved Mover) (Stres 2018, s.v. "Beauty").

In the Middle Ages, the relationship between the beauty of things and their unity is established in a new way. Thus, for Albert the Great (c. 1193–1280), beauty is the "splendor formae", the splendor of the figure. Everything that exists, that is, is composed of form (character) and matter (substance). "Creatures express the content of their character, which is at once the depth and the content of being itself, of their unity, reality and

goodness. This is what gives them beauty. This paves the way for the characteristic medieval scholastic doctrine that beauty belongs to the qualities of being; it belongs to the so-called transcendental qualities of being" (Stres 2018, s.v. "Beauty"). This means that every being, in so far as it is a being, is beautiful. Everything that is, that exists, is beautiful; beauty is proportional to being itself. But beauty is something primordial, something that cannot be faithful to goodness or reality, which are transcendental qualities of being.

## 3. Aesthetic and Transcendental Concepts of Beauty

Thus, in examining the concept of beauty throughout the history of philosophical thought, we encounter at least two aspects, the ontological and the aesthetic:

- The ontological viewpoint conceives of beauty as an ontological, transcendental characteristic of beings, associates it with being itself, with existence itself, and sees beauty everywhere, in all that is, that is being.
- The aesthetic point of view limits beauty to the characteristics of those things that are made to be beautiful, to give aesthetic pleasure, and this is the specific meaning and purpose of art (Stres 2018, s.v. "Beauty" in Lexicon of Philosophy).

### 3.1. The Aesthetic Concept of Beauty

Is the concept of beauty connected with emotions or feelings? Is something subjective or objective? In modern philosophy, the notion of beauty is narrowed. Aesthetics[3] (a term introduced by A. G. Baumgarten, 1714–1762), as a philosophical discipline, has beauty as its object of study. Baumgarten divides cognition into (1) higher or mental cognition (superior); the science that deals with this area is logic and (2) lower or sensual cognition (inferior), which forms the basis of aesthetics and in which the beholding of the beautiful and the enjoyment of the beautiful also take place. In this way, beauty as such is narrowed or reduced to the realm of sensuous-non-temporal perception. This leads to limitations of the beautiful, namely:

(a) Aesthetics is limited to sensory experience (aesthesis). Beauty is opposed to the human being as an object perceived through the senses (visual sense—seeing).
(b) Beauty is limited to reception (receptivity), to passivity, and to the gaze of the observer who is subject to and observes a particular object.
(c) Sense perception is understood subjectively; that is, the beautiful is that which depends on the experience of the senses and liking (taste) each time (Klun 2002, p. 101).

Immanuel Kant broke with the tradition of conceiving the beautiful in relation to other transcendental qualities (*unum, verum, bonum*) and placed the perception of the beautiful within the framework of human perception, the perception and judgment of the beautiful. The beautiful is not in things themselves, but in a human's capacity for rational evaluation, in the domain of the activity of human reason. Beauty is not in being as such, but only in the experience of the subject and in judgment (Klun 2002, p. 101; Stres 2018, s.v. "Beauty"). In his *Critique of the Power of Judgment (Analytic of the Capacity for Aesthetic Judgment)* Kant gives four elements of beauty and the judgment of the beautiful, from which the specificity and characteristic of the aesthetic experience and evaluation of the beautiful can be seen (Stres 2018, s.v. "Beauty"): (1) The beautiful is a matter of taste; the evaluation or appraisal must be free of interest, free of the search for benefit. (2) Beauty is non-conceptual (one does not need to have a concept of the object to experience the beautiful). (3) The beautiful is that which pleases the common, without concept (even if the experience of the beautiful is non-conceptual, the ocean of the beautiful is nevertheless universal, common. (4) The pleasure or pleasurableness experienced in experiencing the beautiful is characterized primarily by what it is, without any other aim or goal to serve. What is beautiful is what deserves to be, without having any other purpose or goal. The essence of beauty, according to Kant, is thus in purposiveness without a purpose (Stres 2018, s.v. "Beauty"). Beauty is not something that has some special goal; beauty is in the experience of a subject and a judgement or evaluation.

*3.2. The Transcendental Concept of Beauty*

If beauty is to be understood as a transcendental quality, it is necessary to go beyond the aesthetics of the modern age. Beauty is grounded in being. Being, insofar as it is being, reveals the beauty of being. The existent "reveals" beyond its image (form, essence, shapeliness, relations) the beauty of being. This glimpse announces itself as the groundlessness, the inexhaustible depth of being, as the splendor of the miracle of being (Klun 2002, pp. 103–4). Every being, insofar as it is being, merely because it is, because it exists, because it is given to exist or to be, is at the same time already beautiful.

Transcendental Beauty and Its Uniqueness

Beauty as a transcendental quality has rarely been treated independently in the history of philosophical thought. Scholastic philosophy did not count beauty among the original transcendental qualities of being (*unum, verum, bonum*), but beauty is always an expression of the good.

Beauty is an expression of the good. The link between the beautiful and the good comes from the Greek tradition (Plato in particular). In this context, the task or mission of beauty is the moral purification of humans. Beauty is thus characterized by order and moderation. This view is carried over into the Middle Ages (Dionysius the Areopagite equates the beautiful and the good).

Beauty is the "splendor of truth". In modern aesthetics, beauty is often interpreted as a "phenomenon" of truth (J.G. Herder 1744–1803). Hegel conceives of beauty as the sensual "radiance of an idea". In a different context, Heidegger conceives of beauty as the revelation of the "truth of being".

Beauty is the perfect unity of all transcendentals. Thomas Aquinas, who after Dionysius the Areopagite links the good and the beautiful (Stres 2018, s.v. "Beauty"), in this context conceives of beauty as something subjective, something that pleases when we see that "something", but not only in the sense of "bodily" seeing, or seeing with the eyes, but in the sense of cognition, which requires harmony. In this context, Thomas remains within the framework of the ancient tradition according to which the beautiful is something that is perfect, harmonious, radiant in its expressive power, and therefore easier to perceive. St. Thomas, also referring to his teacher Albert the Great, describes beauty as "perfectio" (perfection) and "integritas" (integrity, purity, especially in a moral sense), as "debita proportio" (complex proportion) or "consonantia" (harmony, consonance), and as "claritas" (clarity, radiance) (Sth I, 39, 8). The beautiful is thus first of all bound to the good, *bonum* (as perfection), but it is not identical with it, because it is directed to the cognitive power, and is thus intrinsically linked to truth (*verum*). The beautiful is agreeable to cognition; it pleases it. Thomas Aquinas writes that we consider beautiful that which pleases when we look at it (*pulchra enim dicuntur quae visa placent*) (Klun 2002, pp. 103–4).

Beauty is not only in perfection, in completion; it can also be in the draft, the unfinished, the incomplete. In the power of original beauty, every being is beautiful, which suggests that the definition of beauty goes beyond the definition itself in terms of order, symmetry, harmony, and clarity. It is essential to beauty that it is grounded not in the form (formativity) of essence, but in being as such. Beauty is also in feeling or sensation; beauty is the sensation of the overflowing fullness of being. It is a wholeness of feeling that can be metaphorically linked to the heart. Beauty reaches us "to the heart;" it inspires us, attracts us, pleases us, delights us, calls us, and can transform us. The Renaissance thinker M. Ficino (1433–1499) argued that the soul's response to the call of beauty is love. Love is beauty's longing for love: "*amor nihil aliud est, quam pulchritudinis desiderium*" (Klun 2002, pp. 104–5).

## 4. From a Philosophical to a Theological Conception of Beauty

How can one understand the philosophical concept of beauty in theological terms? The completeness and perfection of beauty, which is given to humans to see, to experience or to be "enchanted" by (on the ontological and aesthetic level), is possible only as a reflection of its origin, of the One who is Beauty itself, God. The beauty of experiencing this divine

beauty also lies in the possibility of speaking about it; when you are touched by beauty, you cannot keep it to yourself, but want to share it with others. Beauty happens in relationship, in dialogue, in communication. Pope Benedict XVI defined beauty in relation to liturgy: "True beauty is the love of God, made definitively manifest in the Paschal Mystery. The beauty of liturgy is part of this mystery. It is the highest expression of God's glory and, in a sense, the revelation of heaven on earth" (Benedict XVI 2007, No. 35). Beauty is being in relationship with God, like Benedict said: "*There is nothing more beautiful than to know Him and to speak to others of our friendship with Him*" (Benedict XVI, Homily at the Mass for the inauguration of his Pontificate, 24 April 2005).

Beauty is in itself such that it moves; it leaves no one indifferent. It is not a disturbance that divides, but an excitement that permeates, connects and creates something new; one might even say it changes a person from the inside. In the true sense of the word, only a person can be disturbed, and this opens up a great possibility for humans: to enter into a relationship with another person, into a dialogue, into a friendly relationship, up to the highest level of love, which ends in devotion to the other, in martyrdom (Turnšek 1997, p. 205), or as it is written in the Bible, "*Greater love has no one than this, that one lay down his life for his friends*" (John 15: 13). The beauty of Jesus Christ is the ultimate beauty that binds, heals wounds, and redeems, that attracts with such force that whoever is touched by it can no longer escape it; he is absorbed in this beauty and "surrenders" to it, to Him. As Augustine wrote, "O eternal truth and true love and beloved eternity! You are my God, to you I sigh night and day. When I first knew you, you lifted me up to yourself. . ./. . ./Why your strong radiance, poured out upon me, blinded and repelled my feeble eye, and I trembled in love. . ." (Avguštin 1991, Izpovedi/Confessiones, 7; 10). And he continues, "I loved you late, Beauty, eternally ancient, eternally new, I loved you late!. . ./. . ./. . .You called and called—and you broke through my deafness. . ./. . ./. . .You flashed and glowed—and you drove away my blindness. You breathed sweetly on me, and I drank in your scent—I crave you. I tasted you—and I hunger and thirst for you. You touched me—and I hungered for your peace" (Avguštin 1991, Izpovedi/Confessiones, 10; 27). The beauty that Augustine longs for is the beauty of God Himself, a beauty that is more disturbing than any beauty, for "any beauty" is but a reflection of that eternal and perfect beauty which coincides with the mystery of Goodness and Truth and is finally united in the mystery of Love.

The Way of Beauty is mysterious, often expressing itself in symbolic language, images, and material and immaterial culture, triggering in the human an experience that is not in his power. Beauty is the way that leads to God and the way that God addresses man. As John Paul II writes in his *Letter to the Artists*, beauty is the key to the mystery and a call to transcendence, an invitation to savor life and an invitation to dream of the future. All created beauty is but a reflection of its origin, a reflection of God's beauty, and therefore the beauty of created things can never fully satisfy. The beauty of created things arouses a mysterious homesickness (nostalgia) for God, which is why John Paul II invites artists from all over the world to lead by their creativity to the infinite Ocean of Beauty, where wonder is transformed into holy reverence, into enthusiasm and joy (CD 82, 24–25). God's beauty is irreducible, or as Hans Urs von Balthasar puts it, "Man cannot of himself capture God's glory in a finite figure. . ." (von Balthasar 2008, p. 343). However, since every created being is a reflection of God, and can only analogically correspond to the absolute being of God himself, he can grasp the one, the true, the good, and the beautiful (as the transcendental qualities of being are expressed) even if in different degrees and forms of manifestation. The beauty of the world does not manifest itself except as limited by a finite being or by the harmonious arrangement of finite beings from one to another, but the beauty of God always shines forth in the total otherness of its indivisible magnificence, which transcends and pervades everything (von Balthasar 2008, pp. 341–42).

One of the ways of beauty that speaks to humans in a special way is the beauty of Mother of Christ which is one form of the revelations of God's beauty. Different aspects of Mary's beauty are presented and discussed in the next section.

## 5. *Via Pulchritudinis* and Aesthetic Views of the Virgin Mary

The linking of theological messages with art and sacred images has a long tradition in Christianity. For centuries, through sacred images, believers have discovered religious revelation, found encouragement for their daily lives, and received hope and consolation in social and personal hardships. This is also of paramount importance for today, when in a secular society many people face difficulties in accepting the Church's teaching and morals. The *via pulchritudinis*, or Way of Beauty, is a privileged space for addressing contemporary humans, which includes admiration for the beauty of creation, art, and Christ as the model of holiness (Pontifical Council for Culture 2006; Słupek 2021, pp. 302–3). The sacred images of Mary are an integral part of the *via pulchritudinis* (along with music, songs, architecture, film, etc.). This understanding of sacred images influenced the notion of beauty in Christian art, which Pope Paul VI articulated on 16 May 1975 when he challenged the notion of conflating *via pulchritudinis* with the beauty of Mary (Paul VI 1975). In his address to the participants at the Marian Congress in Rome, the Pope highlighted the twofold approach to Mary. First, he highlighted the so-called Way of Truth (*via veritatis*), which is based on the study of biblical, historical, and theological reflections on Mary. This is the way of theologians and Mariologists (Roten 1998, p. 109). The second approach is the so-called Way of Beauty (*via pulchritudinis*), which can be accessed by every believer. He believed that the *via veritatis* leads to the *via pulchritudinis*, which reveals Mary as the most beautiful being and the ideal of divine perfection. The Pope links the notion of Mary's beauty with the Spirit, who establishes a harmony between human and divine beauty (Paul VI 1975, p. 494). This is expressed in a special way in the Catechism of the Catholic Church, which describes the meaning and role of Mary in symbolic language. Thus, it is written:

> *After speaking of the Church, her origin, mission, and destiny, we can find no better way to conclude than by looking to Mary. In her we contemplate what the Church already is in her mystery on her own "pilgrimage of faith", and what she will be in the homeland at the end of her journey. There, "in the glory of the Most Holy and Undivided Trinity", "in the communion of all the saints", the Church is awaited by the one she venerates as Mother of her Lord and as her own mother. In the meantime the Mother of Jesus, in the glory which she possesses in body and soul in heaven, is the image and beginning of the Church as it is to be perfected in the world to come. Likewise, she shines forth on earth until the day of the Lord shall come, a sign of certain hope and comfort to the pilgrim People of God.* (Catechism of the Catholic Church 2000, p. 972)

Mary's beauty is manifested in the unveiling of the Church. In the dogmatic Constitution on the Church *Lumen Gentium*, adopted at the Second Vatican Council, the Council Fathers wrote,

> *But while in the most holy Virgin the Church has already reached that perfection whereby she is without spot or wrinkle, the followers of Christ still strive to increase in holiness by conquering sin. And so they turn their eyes to Mary, who shines forth to the whole community of the elect as the model of virtues. Piously meditating on her and contemplating her in the light of the Word made man, the Church with reverence enters more intimately into the great mystery of the Incarnation and becomes more and more like her Spouse.* (Second Vatican Council 1964, p. 65)

To understand the beauty of Mary, it is important to emphasize the link between divine beauty and goodness. Hans Urs von Balthasar does not place the beauty of Mary in the context of theo-aesthetics, but in the context of theo-dramatics. In this context, Mary's beauty is manifested in the revelation of God's goodness towards the world and humanity. The encounter of infinite divine and finite human freedom is revealed in the goodness that is the foundation of beauty. von Balthasar says that beauty is where the divine and the human meet. The most perfect beauty seen in this type of encounter is revealed in Jesus Christ himself (von Balthasar 1974, pp. 164–70). Similarly, and not in the same perfection as in Jesus, beauty, and thus the goodness of God, is revealed in the beauty of Mary. In Catholic theology, Mary is considered to express an aesthetic reality and to be the symbol

of the new covenant between man and God, and thus the origin of Christianity. Likewise, Mary's beauty is that she is an eschatological icon of Christian perfection. Mary's beauty is expressed in her promise and hope and the perfect presence of God's love in this world. In a special way, this is manifested in service and mission and in the constant acceptance of God's will. Mary is the place of God's revelation when she bore and nourished God's Son, Jesus Christ. Mary is also considered to be the mediator between humanity and God. It is not unusual, therefore, that it is through the images of the Mother of God and her life story that believers have discovered the meaning of revelation for their daily Christian life. Her images (paintings, statues) and songs have become one of the most successful ways in the Christian tradition to communicate God's revelation and to awaken and strengthen personal faith. At a time when the Catholic Church in Western postmodern society is facing secularization and a decline in faithfulness, the *via pulchritudinis*, which reveals the beauty of Mary, can be one of the best ways of the so-called new evangelization and new catechesis. It must be stressed, however, that the beauty of Mary reveals not only salvific messages but also concrete existential and life experiences that are lived by human in his/her daily life. Thus, the notion of the beauty of Mary is the name for the connection between both Mary's and the viewer's (believer's) experience of life, suffering, longing and divinity.

## 6. Catechetical-Existential Aspects of the Beauty of Mary

One of the best catechetical tools for communicating the salvific message is precisely the approach of Mary's beauty. The reason is that her beauty is manifested in a holiness that is bound not only to divine revelation but also to all the fundamental existential experiences of humans in which the catechized are addressed and challenged. It is a transcendence of the *philosophia perennis*, which is bound up with the fullness of light and grandeur, harmony, and proportion and form as formulated by Plato, St. Thomas Aquinas, and St. Albert the Great. It is the outward image of the inner connection between the life of Mary and the daily life of the human, who recognizes in the successes, trials, and sufferings of life the elements of Mary's life and her experiences. Beauty in this context is not only the awakening of feelings of pleasure and admiration, but the establishment of a relationship between Mary and the believer, and thus a doorway to transcendence and a personal relationship with God. In this context, images of Mary are an excellent means of communicating faith and catechesis. In different life situations and theological messages, the images of Mary appeal to the believer (child and adult) for the transcendent and the eternal. In the Catholic and Orthodox traditions, we distinguish different images of Mary which, through their aesthetics, open the soul of the believer to the supernatural. Why are there different images of Mary? Because the beauty of Mary speaks to human beings in different existential situations. The aesthetic value of Mary is not only in the sense of beauty as such, but in the various existential dimensions (beauty that brings joy, beauty that comforts, beauty that brings hope, etc.) and in her ability to help believers, on the basis of their existential and religious experiences, to understand and accept the various existential experiences and through them to enter into a relationship with God. The aim or goal of Mary's beauty is not Mary, still less the admiration of her beauty in itself, but beauty as a path that directs the gaze beyond her to the transcendent, to the eternal beauty, to God. The foundation of Mary's beauty is her sharing in God's beauty. Such an understanding of Mary's beauty makes it possible to communicate the faith in pastoral care and catechesis. Concretely, this means that pastoral workers can use existential images from the life of Mary, relate them to the life experiences of the believer (e.g., birth, family, peace, suffering, death, resurrection, etc.) and connect them with the Divine. The role of art is precisely to open people up to religious experiences. Pope John Paul II in his Letter to the Artists highlights the purpose and role of art, because art, by its tangibility, is meant to be a bridge to religious experience. Art captures the invisible and makes it visible and perceptible, in other words, art can be faith made concrete, as has been demonstrated by the many artists who have captured the content of the Gospel and the Word in their work (John Paul II 1999, p. 10). The following are some of the artistic images of the beauty of Mary that link the life

and religious experiences of Mary's life with the existential and religious experiences of the believers. These links are the foundation of the catechetical approach of Mary's beauty, awakening the believer's awareness of both the existential and religious events of Mary's life and the existential events of humanity, and connecting them to God.

### 6.1. The Immaculate Mother of God and the Mystery of the Human Conception

In 1613, El Greco completed his depiction of Mary Immaculate, which reveals the Catholic religious truth that Mary was spared original sin at her conception. El Greco, together with other painters such as Peter Paul Rubens, Diego Velázquez (and many other artists), presented to the faithful or viewers one of the religious truths proclaimed by Pope Pius IX in 1854. This image radiates not only the so-called *via veritas* and *via fides*, but also the beauty of Mary, which is manifested in the fact that it opens the heart of the viewer not only to the event of the conception of the Virgin Mary, but also to his own conception as a great mystery and gift of God. Mary's beauty is manifested in awakening the believer's awareness of his own existential dimensions and connecting them to God's action. In the same way, Mary's Immaculate Conception opens the believer's gaze to the mystery of original sin, understood in Catholic tradition as the absence of original holiness and righteousness and in human's innate inclination to evil. Looking at the image of Mary's Immaculate Conception, the Christian can recall the meaning of Baptism, which is precisely the erasure of the consequences of original sin and the opening of the believer's soul to a relationship with God. All the artists mentioned above, such as El Greco, Rubens, Velazquez, etc., in contemplating the life of Mary and her mysteries (e.g., the Immaculate Conception), consciously or unconsciously discovered human, and thus their own, existential dimensions and gave them a divine connotation and meaning. The catechetical approach of the beauty of Mary comes to a particular expression when the believer becomes aware of all the existential and religious dimensions of the Immaculate Conception and relates them to his own life. Reflection on the event of the Immaculate Conception can be a framework in the catechesis both to meditate on God's miracle and to encourage reflection on the conception of every human being and his or her relationship to God. The mention of well-known artistic depictions of the Immaculate Conception of Mary is an opportunity for pastoral workers to use works of art by world-famous artists to convey the above-mentioned religious message. Similarly, works of art by well-known artists show that the Virgin Mary's conception was so important to the experience of faith that they devoted their thinking and work to this aspect of Mary's beauty. These works of art can be used as pastoral tools to awaken awareness of one's own conception and the religious mystery of Mary's conception.

### 6.2. The Archetypal Image of Mary with Child and the Meaning of Motherhood

Motherhood is a symbol familiar to all major religions and cultures. In Christianity, Mary is inextricably linked with motherhood. This linking has its biblical origins in the event of the incarnation and birth of Jesus and in the event on the cross when Jesus gives his mother Mary to the Apostle John. In this event, Christian tradition recognizes the message that Jesus gave Mary to the Apostles and to all believers to be their mother. Her motherhood is manifested in her unconditional love and devotion to her child and her willingness to renounce her own interests time and again. Christian theology is about obedience to God, renouncing one's own will and accepting God's will. Mary subordinated her life plans to God's and accepted God's will by becoming God's mother. The image of Mary with the child is one of the most powerful symbols of Christianity and of God's revelation. It represents the unbreakable covenant between God and humanity with Mary as the representative of humanity and Jesus as the Son of God. This image is a symbol of the Incarnation, which anticipates the redemption of humanity. God recognizes himself in the poorest of the poor and God recognizes himself in the least of the poor. It is a living testament of love that marks the covenant between God and humanity. Mary represents the communion of saints and the so-called feminine aspect of creation. For the believer, the sight

of Mary with child connects with the experience of motherhood that she has experienced or is still experiencing in her or his family. Therefore, in the catechetical field, the image of Mary holding Jesus is an excellent means of awakening both human and religious messages. In the same way, Mary's beauty is a means of communicating universal archetypal qualities and virtues such as fertility, nourishment, love, and compassion. From a catechetical perspective, the artistic depictions of Mary with her child are a starting point for exploring all themes related to motherhood, especially the experience of motherhood in the believer's own family and, for example, how the believer experiences Mary's motherhood during the period of mourning for her deceased mother. It is also through Mary's beauty that the believer discovers how God cares for each person in a motherly way. Experience shows that the faithful understand Mary not only as the mother of Jesus, but as their spiritual mother. Just as Jesus, on the cross, gave his mother Mary to the apostle John with the words, "Behold, your mother" (John 19:27). And John accepted Mary as his (spiritual) mother.

### 6.3. *The Sorrowful Mother of God, Miraculous Images, and the Existential Experience of Suffering*

Mary is often depicted in popular devotions as the suffering mother. She is most often depicted by artists with a pierced heart in which one or seven swords are planted. The experience of Mary's suffering is the subject of many devotions and venerations. In the Catholic Church, a special liturgical feast is dedicated to this experience. It reminds the faithful of the suffering that Mary experienced at Joseph's misunderstanding when she became pregnant, Joseph's premature death, and the suffering and death of her son. Mary is the woman who experienced the various forms of suffering with which the believer can identify. She is an image of patience and resignation, a source of spiritual grace and strength, and one who assures the presence of God in the afflictions and sufferings of the believer. It is not unusual, therefore, that Mary is the patroness of the suffering and the sick, as is especially highlighted in the litanies. Mary's beauty, which combines her suffering and her resignation to God's will, can be a great inspiration, especially to the believer who is faced with suffering and despair. In their catechetical work, pastoral workers can use images of Mary's suffering to talk about acceptance and the meaning of human suffering. By evoking the memory of the believer's suffering (e.g., illness or accident) and making connections with the suffering of the Virgin Mary and Jesus (God) pastoral workers can express the deeply Christian religious message that all suffering and death is ultimately followed by resurrection and eternal life with God.

### 6.4. *Virgin Mary, Queen of Families*

The family is a value that holds a special place in all cultures and religions. Mary, together with Joseph and Jesus, is the image of the family, which can be the image and archetype of every family. The Holy Family conveys the message that there are no ideal families. There are only concrete families in this world, with challenges, longings, trials and mysteries. In the viewer of the image of Mary, Queen of the Family, the concept of family is linked to his or her experience of family, which can be positive or negative. Christians commend themselves to Mary, Queen of the Family, because she herself has experienced all the dimensions of family life and understands the family's hardships and the joys of the ordinary person. Mary's beauty or aesthetics is reflected in Mary as Queen of the family, who awakens gratitude and joy in the heart of the human for his own family and invites him to ask for a solution to the difficulties that people face in their own families. Pastoral workers can use images of the Virgin Mary as Queen of the Family to talk about the importance of family life for Christians in catechesis. They also bring to the fore the various expressions of family love, solidarity, difficulties and sufferings that the faithful face in their own families. They can also raise the issue of non-ideal families such as the divorced, widowed, infertile, etc., and the meaning of family forgiveness and prayer.

### 6.5. Mary of the Assumption and Eschatological Hope

The image of Mary of the Assumption represents a religious truth in the Catholic Church that has evolved over the centuries and was proclaimed by Pope Pius XII in 1950 (Pius XII 1950). This religious dogma states that Mary was spared death and that she was assumed into heaven, that is, she was taken up to heaven with her soul and body. Such an image of Mary reminds the believer of the existential experience of the transience of one's own bodily life, the reality of death and eternity. Mary of the Assumption has a special place in heaven and is considered the intercessor to whom Christians entrust their eschatological future. The catechetical value of Mary's beauty especially comes to the fore when she urges the believer to prepare for the last events of her or his own life and to remember the deceased whom he hopes are already with God. In general, images of the Assumption can be used in catechesis to address topics such as preparing for death, accompanying the dying, different types of burial, ethical topics such as ageing, suicide and euthanasia, and religious eschatological truths such as vice, heaven, and hell. The image of Mary of the Assumption is an excellent catechetical tool to prepare the believer for the most important experiences of life and spirituality.

### 6.6. Mary as Dignified Woman

The images of Mary show a woman who has fulfilled her feminine mission. In modern times, we speak of the dignity that every woman enjoys, which is inseparable from women's equality. The beauty of Mary tells us that all women are created in the image of God and that they have the same dignity as men. It was also Mary who played a key role in the incarnation of God and witnessed the resurrection of Jesus Christ. Through her life mission, Mary expressed the virtues and Christian perfection that came to expression especially in her acceptance of God's will and of the Cross. In this, Mary's beauty is a source of inspiration for all women who wish to affirm their femininity and for men to become aware of the dignity and respect of women. Mary's beauty also conveys the message that every woman in the Church has an important mission in prayer, in following Christ, and in works of love, as Pope John Paul II stated in his apostolic letter *Mulieris dignitatem* (John Paul II 1988, pp. 28–30). Pastoral workers can encourage the faithful to reflect on the dignity and social role of women by presenting artistic images of Mary. They can also point out actions and circumstances that oppose respect for women (e.g., prostitution, lower wages, discrimination, violence, etc.). They can also highlight figures of women who have made an important mark on human history (e.g., in art, science, and politics). Last but not least, the image of Mary is always an opportunity for pastoral workers to talk about femininity, the differences between men and women and the positive role that women play in the Church.

### 6.7. The Images of Marian Apparitions as God's Presence in the World

The many Church-recognized apparitions and representations of Mary tell the Catholic faithful that God needs Mary and that through her He is present in the world and in the life of the believer. The apparitions at Fatima, Lourdes and Guadalupe are of great importance for daily Christian life and for the development of various devotions. Often Mary's messages are linked to the call to prayer, penance, reconciliation, and commitment to peace. In such messages, the beauty of Mary is indeed a source of encouragement for the Christian mission. In her beauty, they recognize that God is present in this world and that life is meaningful when we are doing God's will. Pastoral workers can present local and the world's major Marian shrines and the importance of pilgrimages in a catechesis. They can also present the messages of Mary or God in the shrines and their universal call to prayer, penance and peace.

### 7. Conclusions

Beauty expressed through works of art can really open the eyes of the mind and heart and direct the human spirit to transcendence. The beauty of art awakens inner emotionality, evokes elation in silence and leads to a coming out of oneself. The concept of beauty

could be presented from two aspects: transcendental and aesthetic. Both are revealing the beauty of its source, divine beauty, beauty of God. The Mother of God, Mary, as a reflection and carrier of beauty, with different perspectives of her life, addresses the modern human and leads him/her to the highest beauty, to God. The meaning of Mary's beauty, revealed through the holy images of Mary, lies in the believing Christian's identification with the divine. Such an approach is often subjective, since it is based on the experience of the person who looks at the images of Mary and, in looking at Mary, consciously or subconsciously discovers the existential dimensions of both Mary's life and his own. In doing so, he also discovers theological and eschatological messages. In this way, the beauty of Mary can be an excellent catechetical tool, addressing humans not only intellectually, but also existentially and emotionally. Of course, such an address is not without its dangers. Such an understanding of the beauty of Mary can lead to a reduction of Mary to the above-mentioned messages alone, ignoring the so-called historical image of Mary, which was embedded in the culture and historical moment in which she lived. Nevertheless, the beauty of Mary is the concept of a so-called "descending theology", which is intimately discovered and experienced by people. For this reason, the beauty of Mary is part of the *via* pulchritudinis, since it reveals not only Mary, but also the fundamental dimensions of the life of the believer and the main messages of the faith. This is why the beauty of Mary is one of the best forms of catechesis, because in this catechetical approach there is no difference between so-called sitting and kneeling theology, just as there is no difference between theology and spirituality. Faith is an integral part of the theological beauty of Mary. Therefore, the beauty of Mary is always a process of reconciliation between the so-called Mary of history, Mary of doctrine, and Mary of faith.

**Author Contributions:** Writing—original draft, M.P.R. and T.S. All authors have read and agreed to the published version of the manuscript.

**Funding:** This research received no external funding.

**Institutional Review Board Statement:** Not applicable.

**Informed Consent Statement:** Not applicable.

**Data Availability Statement:** Not applicable.

**Conflicts of Interest:** The authors declare no conflict of interest.

## Notes

[1] Hippias, as a universal thinker, convinced that he was even wiser than the "seven wise men", gave a lecture in Sparta "on the fine, beautiful chores" and wanted to speak about this also in Athens, so Socrates interrogated him and tested his true knowledge (Kocjančič 2004, p. 916).

[2] The Greek concept *kalón/beauty* does not fully correspond to our modern concept (Kocjančič 2004, p. 916).

[3] Aesthetics—the term comes originally from the Greek world for sense-perception and is concerned with our perception of beauty, whether in art or nature and more specifically with the nature of art and our human response to it (Cottingham 2008, p. 694).

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
