# Peer review of "The Question of Beauty and the Aesthetic Value of the Image of the Mother of God in Pastoral Care and Catechesis"

_religions, doi:10.3390/rel15010101_

Round 1

Reviewer 1 Report

Comments and Suggestions for Authors Surprisingly, there is no mention of the aesthetic value of the image of the mother of God in pastoral care and catechesis anywhere in this paper, despite the title.

This paper needs to be rewritten focused on the value of the image of the Mother of God in pastoral care and catechesis. That value is clearly not an aesthetic value. Otherwise, a justification needs to be provided from the outset as to how aesthetic values relate to pastoral care and to catechesis.

Further, there is no basic review of any literature to build on what has been said already. But what is said here is very familiar and easily accessed elsewhere.

A lot of pious statements about Mary and the Church do not give credibility to a promised subject matter that the reader is constantly looking for, namely, the aesthetic value of the image of the Mother of God in pastoral care and catechesis.

Many statements are unsubstantiated. Where does Pope Paul link “the notion of Mary's beauty with the Spirit, who establishes a harmony between human and divine beauty.” These kinds of statements need to be supported. Instead, the author refers to the Catholic Catechism presumably authored by Pope John Paul II.

What has obedience to God, renouncing one's own will and accepting God's will got to do with the archetypal image of Mary with child and the meaning of motherhood? The connections need to be made and they need to be clear.

How does the image of Mary with the child represent the unbreakable covenant between God and humanity? Which covenant? Again, so many unsubstantiated statements. How does El Greco’s depiction of Mary Immaculate “radiate not only the so-called via veritas and via fides, but also the beauty of Mary, which is manifested in the fact that it opens the heart of the viewer not only to the event of the conception of the Virgin Mary, but also to his own conception as a great mystery and gift of God”? But how?  Only in a learned Catholic tradition would any of this make sense.

Further, how does El Greco’s depiction of Mary Immaculate reveal Mary conceived without sin more than any other version of this theme? Why El Greco? Why Rubens and Velázquez only? What is special about these artists and their masterpieces? The reader is left wondering.

What does “the catechetical approach of the beauty of Mary” mean? Why is this not explained? The reader has to presume that the author is relating the beauty of Mary to religious instruction given in preparation for Christian baptism or confirmation, or to religious teaching by means of questions and answers. Despite mentioning this approach, no attempt is made to provide a catechetical approach, whatever this means for the author.

The biggest disappointment is that the application to pastoral care, which would be noteworthy, is missing. Again, this is claimed or promised but never given. In my view, this would be where the paper’s strength ought to lie. But again it is not provided. This would also need to be reinforced from experts within the fields of catechetics and pastoral care.

Further, it is not clear to me what images the author is meaning when they are mentioned. No reference to specific images is given in 5.5 and 5.6, just the titles of Marian themes such as the assumption and mother and child. Is the reader expected to imagine a general image? Is the use of the word “image” a mental image? This is not clear. Comments on the Quality of English Language

The quality of English used is fine.

Author Response

Dear Reviewer,

thank you for all your work and useful comments. We have taken them into account. We hope that the article is adequately prepared now.

Kind regards.

Mateja Pevec Rozman, Tadej Strehovec

Reviewer 2 Report

Comments and Suggestions for Authors

My thanks to the author for their detailed scholarship and their investigation of the beauty of Mary in dialogue with some major figures in philosophical and Catholic theological aesthetics. While this article requires revision, I want to make sure to praise this central idea to desire for reflection on links between the beauty of Mary to the tradition of aesthetics.

One area that requires substantial revision regards the structure of the article. Primarily, the two sections are not linked. On my reading, I am not entirely sure how the preparation review of aesthetics helped illuminate the work with the image of Mary along the via pulchritudinis. I recommend that each locus in section 5 make an explicit and direct reference to one of the authors referenced in the history of philosophy.

There are also two problems with the summary of the history of aesthetics. The first is its overreliance on dictionaries and reference texts. Why does this article summarize general knowledge about the history of aesthetics? I enjoyed reading this summary; it is a well written “general introduction” to this version of the aesthetics story. But the author needs to make it abundantly clear why and how this approach to (European / Western) aesthetics supports the arugment about Mary.

I recommend that the author reconcieve of the article as an articulation of this phrase as its central thesis: “the notion of the beauty of Mary is the name for the connection between both Mary's and the viewer's (believer's) experience of life, suffering and longing” (ll. 326-327). Life, suffering, and longing would be fascinating angles to tell the story of aesthetics. Then the author could use these interpretive keys in order to turn to the image of Mary as an expression of that beauty. This would also help contextualize some of the references.

When it comes to Marian images, the paper needs to provide more on each locus. This is the section of actual originality, but each image remains illustrative without further argument. I suggest connecting each of these “aspects of the beauty of Mary” explicitly to a key theme from the review of aesthetics. I further suggest that the author either clearly link these images to studies particular works of art (as in the passing reference to El Greco) or address the image of Mary as an archetype of Beauty. In either case, the author will need to offer more theological interpretation and commentary. As it stands, the article does not make an argument but merely identifies the catechetical possibility of Marian imagery. These are elegant sermon illustrations, but the article does not present itself as a manual for preaching in light of the beauty of Mary. These studies of images of Mary are also devoid of citations, and I recommend that the author match the careful scholarship of the aesthetics sections with careful scholarship in the Mariological readings.

At line 420, be careful not to equate “feminism” (a theoretical framework) with “feminine mission.” This is an understandible linguistic mistake, but the paragraph displays ignorance of feminist theology where it promises “Mary’s feminism.” A better title might be “Mary as Dignified Woman.”

Finally, a point of theological and scholarly framing: as currently written, the author provides a general history of European aesthetic theory beginning in Plato and Artistotle, and then arrives at its fascinating typology of the image of Mary. The issue, of course, is that this rather particular version of the history of theological aesthetics presents itself as a normative approach. The author does not cite Hans Urs von Balthasar’s Herrlichkeit (Glory of the Lord), the major Catholic theological aesthetics that influenced John Paul II and Benedict XVI. In order for this “summary” to make sense, I find the lack of even a passing refernece to von Balthasar rather glaring. Please do not misunderstand my point: it is not necessary for the author to approach theological aesthetics in Balthasar's way, but certain turns of phrase in the paper’s conclusion (that is, “kneeling theology” in line 463) are Balthasarian. Whether the author avoids von Balthasar by accident or on purpose, I highly recommend reviewing Glory of the Lord Vol. 1 (Ignatius: 1988) as a dialogue partner. There are a number of quite accessible summaries available. A review of von Balthasar will also help the piece to be more in line with contemporary Catholic Mariology.

I wish the author the best of luck in revision of this piece!

Author Response

Dear R

Dear Reviewer,

thank you for your work and all helpful comments. We have taken them into account.
We hope that the article is now adequately prepared for publication.

Kind regards.
Mateja Pevec Rozman and Tadej Strehovec

Round 2

Reviewer 1 Report

Comments and Suggestions for Authors

This is a pious article. I have read the additions on pastoral work and catechesis. Some make sense. Others do not or are poorly written. It is not sufficient to merely add them at the end of paragraphs. They need to be integrated into the manuscript.

At this time, this article does not meet the basic criteria for a scholarly journal. The content remains uncontextualized with respect to its claims. Nearly every statement is unsupported. Most listed references and never cited. There is nothing in this article that cannot be simply obtained very easily at many other places elsewhere on Mary and art.

Again, it not acceptable today to use English terms in the academy of theology that do not speak to an inclusive audience. 

Comments on the Quality of English Language

No comment.

Author Response

Dear reviewer,

thank you for your comments. We are sending our reply in attachement.

Kind regards!

Mateja Pevec Rozman, Tadej Strehovec

Reviewer 2 Report

Comments and Suggestions for Authors

Dear friends,

Thank you for your careful consideration of my review. It is always appreciated when this review process leads to stronger scholarship as it has in this case. 

I find your adjustments more than adequate to meet my general concerns as reviewer. 

I want to strongly encourage the authors to include some of the justification argument presented in their reply to me as part of a new introduction to the text of the article. The article makes much more sense (to me) given the author's position on the history of aesthetics and the role of Mary. They made some very clear points about their perception of the need for their work, and I think those points deserve to be a part of the article itself.

At this point, any further commentary from me would proceed in the mode of scholarly "thinking with" rather than as peer review. 

My thanks to the authors for this contribution.

Author Response

Dear friend,

thank you very much for your valuable comments. As you suggested, we have added an introduction to the article that summarises the main points of the article itself. Thank you again for helping us to improve our article.

Thank you for your work and your valuable comments.

Kind regards.

Mateja Pevec Rozman and Tadej Strehovec
